# Construction of scFv Antibodies against the Outer Loops of the Microsporidium *Nosema bombycis* ATP/ADP-Transporters and Selection of the Fragment Efficiently Inhibiting Parasite Growth

**DOI:** 10.3390/ijms232315307

**Published:** 2022-12-04

**Authors:** Viacheslav V. Dolgikh, Igor V. Senderskiy, Sergej A. Timofeev, Vladimir S. Zhuravlyov, Alexandra V. Dolgikh, Elena V. Seliverstova, Diloram A. Ismatullaeva, Bakhtiyar A. Mirzakhodjaev

**Affiliations:** 1All-Russian Institute of Plant Protection, Podbelsky Chausse 3, 196608 Saint-Petersburg, Russia; 2All-Russia Research Institute for Agricultural Microbiology, Podbelsky Chausse 3, 196608 Saint-Petersburg, Russia; 3Sechenov Institute of Evolutionary Physiology and Biochemistry, Thorez 44, 194223 Saint-Petersburg, Russia; 4Scientific Research Institute of Sericulture, Ipakchi Str. 1, Tashkent 100069, Uzbekistan

**Keywords:** silkworm *Bombyx mori*, microsporidia *Nosema bombycis*, ATP/ADP-transporters, scFv immune library, phage display, Sf9 cells, RT-qPCR, immunoblotting, parasite growth suppression

## Abstract

Traditional sanitation practices remain the main strategy for controlling *Bombyx mori* infections caused by microsporidia *Nosema bombycis*. This actualizes the development of new approaches to increase the silkworm resistance to this parasite. Here, we constructed a mouse scFv library against the outer loops of *N. bombycis* ATP/ADP carriers and selected nine scFv fragments to the transporter, highly expressed in the early stages of the parasite intracellular growth. Expression of selected scFv genes in Sf9 cells, their infection with different ratios of microsporidia spores per insect cell, qPCR analysis of *N. bombycis* PTP2 and *Spodoptera frugiperda* COXI transcripts in 100 infected cultures made it possible to select the scFv fragment most effectively inhibiting the parasite growth. Western blot analysis of 42 infected cultures with Abs against the parasite β-tubulin confirmed its inhibitory efficiency. Since the VL part of this scFv fragment was identified as a human IgG domain retained from the pSEX81 phagemid during library construction, its VH sequence should be a key antigen-recognizing determinant. Along with the further selection of new recombinant Abs, this suggests the searching for its natural mouse VL domain or “camelization” of the VH fragment by introducing cysteine and hydrophilic residues, as well as the randomization of its CDRs.

## 1. Introduction

Microsporidia are a large group of fungi-related obligate intracellular parasites that infect almost all types of animals, including mammals and humans [1]. Being widespread entomopathogens [2,3], microsporidia infect many domestic and beneficial insects [4,5], causing significant economic damage.

Pébrine is a devastating microsporidiosis for the world’s sericulture. This disease of the silkworm *Bombyx mori* is caused by the microsporidium *Nosema bombycis* [6]. Since the first description of pébrine in France in 1845, traditional sanitation practices such as culling of infected insects and eggs, using healthy animals, remain the main strategy to control *N. bombycis* infection [7]. This actualizes the search and development of new approaches to increase the silkworm resistance to *N. bombycis*.

Most of these studies are focused on the search for parasite genes as effective targets for suppression the growth of *N. bombycis* using RNA interference (RNAi) methods [8,9,10,11]. Along with double-stranded RNA (dsRNA) and small RNA (siRNA) molecules used to specifically suppress the expression of microsporidia genes, metal (silver) nanoparticles may also be effective in the treatment of infections caused by *N. bombycis* [12].

As an alternative to the treatment of infected insects with various compounds, their resistance to intracellular parasites may be increased by the infection-triggered biosynthesis of certain molecules that inhibit the microsporidia growth in infected cells. Along with dsRNA molecules that specifically suppress pathogen gene transcription, recombinant single chain antibodies (Abs) (scFv fragments) against important parasite proteins, heterologously expressed in the infected host cell, should also suppress the development of microsporidia. The *N. bombycis* SWP12 spore wall protein was found to be the first target for scFv Abs that suppress pathogen growth in an infected Sf9 cell culture [13]. Recently, two scFv fragments against *N. bombycis* hexokinase were fused to the F-box domain of *Drosophila melanogaster* to ubiquitinylate the parasite’s secretory enzyme for its proteasome degradation and suppress parasite growth [14].

Since the molecular size of scFv fragments is about 30 kDa, it should be sufficient to block the transmembrane channel of parasite transporters without additional modifications of recombinant Abs. One of the most interesting and important microsporidia membrane transporters are plastid-bacterial (non-mitochondrial) ATP/ADP carriers (AACs) [15,16,17], first discovered in parasitic bacteria [18,19] and plant plastids [20]. In the case of microsporidia, these solute transporters with 12 transmembrane domains import host-derived ATP to provide energy for intracellular growth of the pathogen.

Previously, we fused 18 outer hydrophilic loops of three *N. bombycis* AACs (*Nb*AACs) contacting with the cytoplasm of an infected host cell [21]. The chimeric gene and its fragments, encoding loops of individual transporters, were overexpressed in *E. coli* as insoluble inclusion bodies (IBs). In this study, we constructed an scFv immune library against this chimeric antigen *NbAAC1-3*, isolated the first nine recombinant Abs against the *Nb*AAC1 transporter, and selected the most effective inhibitory fragment which was promising for further modifications.

## 2. Results

### 2.1. Constructing of scFv Immune Library, Its Panning and Selection of NbAAC1-Specific Abs

To obtain an antigen for mice immunization, we overexpressed the *Nb*AAC1-3 chimeric protein carrying the outer loops of three *N. bombycis* ATP/ADP transporters [21] in *E. coli* and purified it by SDS-PAGE followed by gel electroelution (Figure 1A). Thoroughly dialyzed fractions containing pure protein were used to immunize mice, and sera were analyzed by immunoblotting with *Nb*AAC1-3 fragments carrying loops of individual transporters (Figure 1B).

A mouse whose serum showed approximately equal levels of polyclonal Abs to the loops of three *N. bombycis* transporters was chosen to construct the scFv immune library.

The representativeness of the constructed library was about 10 million *E. coli* transformants, and we infected 4 × 10^9^ bacterial cells (200 mL of culture with initial OD_600_ 0.025) with helper phage. Thus, the number of bacterial transformants used in the library panning exceeded its diversity by four hundred times. In order not to divide the pool of phages into three parts, we cut out *Nb*AAC1, *Nb*AAC2, and *Nb*AAC3 immobilized on nitrocellulose as different figures for the first round of panning and incubated them with viruses in one tube. The second and third rounds of selection were performed separately for each chimeric protein.

Heterologous expression of 180 selected genes in *E. coli* (60 bacterial transformants for each chimeric protein) followed by dot blot analysis of their antigen-binding capacity allowed us to select 10 *Nb*AAC1-specific scFv variants (Figure 2A). Dot (Figure 2B) and Western (Figure 2C) blotting confirmed the specificity of these recombinant Abs to *Nb*AAC1. At the same time, we did not find any scFv Ab specific for *Nb*AAC2 or *Nb*AAC3 chimeric proteins.

An additional analysis of 96 colonies carrying scFv genes selected against *Nb*AAC2 and the same number of colonies with genes panned against *Nb*AAC3 also did not reveal any producer of recombinant Abs specific to these proteins. Some expressed scFv Abs non-specifically stained all four or three chimeric proteins placed in the corners of nitrocellulose squares (Figure 2A), but none of the examined fragments specifically recognized *Nb*AAC2 or *Nb*AAC3. This result suggests the lower immunogenicity of *Nb*AAC2 and *Nb*AAC3 compared to *Nb*AAC1.

### 2.2. Originality of Selected scFv Sequences

Analysis of the amino acid (a/a) sequences of 10 selected recombinant Abs (Table 1, Figure 3) demonstrated (1) 100% identity of independently selected scFv3 and 10, (2) the same VL part in scFv1 and 5, identified as a human IgG domain from the commercial pSEX81 phagemid that was retained during library construction, (3) no VL domain in scFv4, (4) no identical VL or VH fragments in other scFv Abs. VH domains of scFv4, 7 and 8 showed 91–93% identity, differing only in a few a/a residues. However, the identity of the VL parts of scFv7 and 8 was 81% and this domain was absent in scFv4. Similarly, scFv fragments 3, 6 and 8 showed the greatest similarity of VL domains (93–95% identity), but their VH parts differed more strongly (Table 1). Variable fragments of selected recombinant Abs have 87–93% (VHs) and 91–98% (VLs) identity with the most similar sequences deposited in GenBank. No fragment was completely identical to any of the deposited sequences.

### 2.3. Heterologous Expression of Selected scFv Fragments in Sf9 Cells

To express nine different *Nb*AAC1-specific scFv fragments in insect cells, their genes were cloned into the pIB/V5-His plasmid. Lipofection of Lepidopteran Sf9 cultures with engineered plasmids followed by selection of transformants in the presence of the antibiotic blasticidin S and immunoblotting with anti-c-myc Abs demonstrated heterologous expression of cloned genes in insect cells (Figure 4). To correlate scFv accumulation in the transformants with the total protein concentrations in the samples, we repeated SDS-PAGE with the same aliquots and stained the gel with Coomassie. Since the lowest content of scFv 3, 6 and 8 in transformants correlated with lower protein concentrations in these samples, this generally indicates efficient expression of recombinant Abs in Sf9 cells. At the same time, the studied scFv fragments were differently expressed in the blasticidin-selected transformants, and the concentration of scFv1 significantly exceeded the content of other recombinant Abs. The single VH domain scFv4 showed the smallest molecular weight. The fragments scFv1 and 5 carrying VL part of a human IgG from the pSEX81 phagemid were also smaller than the other Abs.

### 2.4. Expression of NbAAC1-Specific scFv Abs Differently Affected the Parasite Intracellular Growth

To select a recombinant Ab with maximal *N. bombycis* inhibitory activity, nine blasticidin-resistant transformants were infected with 20 microsporidia spores per insect cell, and expression of the *N. bombycis* polar tube protein PTP2 gene was analyzed 7 days after inoculation, using real-time qPCR. To control the quality of RNA isolation and cDNA synthesis, the cutworm *Spodoptera frugiperda* cytochrome c oxidase subunit I (COXI) transcripts were analyzed in the same cDNA samples as a reference. A total of 64 independent infections repeated 4 to 9 times for each scFv variant were analyzed.

As shown in Table 2, the number of *N. bombycis* PTP2 transcripts relative to insect COXI ones varied in cell cultures expressing different scFv fragments. The most efficient development of the parasite was observed in cultures expressing scFv4, 6, and 8. The ΔCq (5.2–6) and *Nb*PTP2 Cq (19.9–20.2) values in these variants were significantly different (*p*-value < 0.05) from the values detected in the cultures expressing six other Abs (6.9–8.5 and 20.9–22.7, respectively). Cloning of PCR products, specifically amplified with primers *Sf*COXI and *Nb*PTP2, constructing of calibration curves and counting of transcripts in analyzed cDNA samples showed that cultures with the lowest (expression of scFv4) and highest (expression of scFv5) ΔCq demonstrated a tenfold difference in the number of *Nb*PTP2 transcripts per thousand of *Sf*COXI ones (11 and 1.1 copies, respectively). The constructing of calibration curves also showed that PCR efficiencies was 92.4% and 122.4% for primers *Sf*COXI and *Nb*PTP2, respectively.

Analyzing these results, we hypothesized that simultaneously effective (ΔCq ≤ 6) and inefficient (ΔCq ≥ 7) growth of microsporidia in infected cultures expressing the same scFv fragment may be due to its low suppressing activity in some experimental replicates, along with poor activation of spores or their aggregation and adhesion in others. Since scFv5, 7 and 9 showed the highest ΔCq values along with inefficient growth of *N. bombycis* (ΔCq ≥ 7) in all independently infected cell cultures (Table 2, right column), they were chosen for the next experiment. Retention of a human IgG VL domain from the commercial phagemid pSEX81 in scFv5 was an interesting result of the experiment, as this recombinant Ab showed the highest *N. bombycis* inhibitory activity. Since scFv fragments 5 and 1 carry the same “artificial” VL domain, the latter was also included in the next experiment.

### 2.5. The Effectiveness of scFv5 to Suppress Intracellular Growth of N. bombycis Was Confirmed by Additional Infections and qPCR Assays

Sf9 transformants expressing the four scFv Abs selected in the first experiment were infected with lower doses of *N. bombycis* spores. In this experiment, we reduced the multiplicity of infection to ten and three spores per insect cell (Table 3).

As expected, the *Sf*COXI Cq values in Table 2 and Table 3 were approximately the same (13.5–15.3 and 13.8–15.5, respectively), and the decrease in infection load was accompanied by an increase in *Nb*PTP2 Cq (19.9–22.7 and 22.1–26.1, respectively). At the same time, we did not observe a clear relationship between a decrease in the multiplicity of infection from ten to three spores per cell and the number of parasite transcripts in cDNA samples. In any case, both infections at a ratio of ten and three spores per insect cell confirmed the highest suppressive activity of scFv5, showing a statistically significant difference in ΔCq values (Table 3). As in the first experiment, scFv5 again demonstrated the highest Cq, ΔCq values and the lowest number of *Nb*PTP2 transcripts per million *Sf*COXI ones.

In the final qPCR experiment, we used another pair of *Nb*PTP2 and *Sf*COXI-specific primers, and also added a control variant of cell cultures transformed with a plasmid carrying the gene encoding eGFP. Cell cultures expressing five scFv fragments and eGFP were infected with 50 microsporidia spores per insect cell, and after 7 days of cultivation, the content of parasite and insect transcripts was analyzed by qPCR with *Sf*COXI-2/*Nb*PTP2-2 primers (Table 4). As in previous experiments, the results of qPCR with a new pair of primers showed that of all of the analyzed variants, scFv5 most effectively inhibits the intracellular growth of the pathogen, demonstrating the highest values of Cq *Nb*PTP2-2 and ΔCq *Sf*COXI-2/*Nb*PTP2-2 (Table 4). The lowest Cq and ΔCq showed scFv8 and these values were similar to those observed for the eGFP-expressing control cultures.

### 2.6. Western Blot Analysis of Infected Transformants with Abs against N. bombycis β-Tubulin Confirmed the Inhibitory Activity of scFv5

Previously, we produced polyclonal Abs against *N. bombycis* β-tubulin overexpressed in *E. coli* and demonstrated the promise of Western blotting for detecting intracellular growth of the parasite in infected Sf9 cell cultures [22]. In this study, we attempted to use immunoblotting with these Abs to further confirm the *N. bombycis*-inhibitory efficacy of scFv5 compared to other selected recombinant Abs. Blasticidin-selected transformants were infected with microsporidia at a ratio of 10, 20 or 50 spores per cell, and after cultivation for 1.5 h, 4 and 7 days, the samples for SDS-PAGE were prepared. According to the results of immunoblotting, parasite β-tubulin was not detected in freshly infected cultures (1.5 h after infection, not shown), but its detectable accumulation was found in 42 samples of 4- and 7-day cell cultures (Figure 5).

When cell cultures were infected with 10 or 20 spores per cell, Abs against *N. bombycis* β-tubulin stained two protein bands. The lower band corresponded to the parasite β-tubulin, and the upper one, to the homologous *S. frugiperda* protein [22]. When cell cultures were infected with 50 spores per cell, the parasite protein band was stained more brightly.

Although visual assessment of protein band staining after immunoblotting is not quantitative, this method provided additional evidence for the high *N. bombycis*-inhibitory activity of scFv5. A comparison of the staining intensity of the parasite β-tubulin in samples of cell cultures infected with three doses of spores on the 4th and 7th day after inoculation showed that in each of these six groups, the lowest growth of the pathogen was observed in the case of scFv5 expression (Figure 5, indicated by arrows).

### 2.7. Identification of Complementarity Determining Regions (CDRs) of the scFv5 Unique VH Domain

A blast analysis of the scFv5 VH domain a/a sequence showed that the most similar fragment deposited in GenBank (accession number 5DO2_C) has 86.9% identity.

The identification of the scFv5 VH CDR1 and CDR2 did not show similarity to corresponding sequences of other selected Abs, except for the VH part of scFv3 (Figure 6). In this case, the identity of CDR1 and CDR2 regions was 78% and 82%, respectively. As expected, the highly heterogeneous CDR3 region in the VH part of scFv5 was significantly different from the corresponding regions of all other selected VH fragments. In GenBank, we found only one VH domain with a CDR3 identical to the same scFv5 region (Figure 6, accession number BCB24250.1), but its CDR1 and CDR2 were significantly different from those of scFv5. The CDR1 and CDR2 of two other VH fragments (Figure 6, AAO19046.1 and CCQ13148.1) with most similar CDR3 were also not identical to scFv5 regions. The unique sequence of the VH domain of *N. bombycis*-inhibitory scFv5 was deposed in GenBank.

## 3. Discussion

To construct an immune library of scFv Abs specific to *Nb*AAC1-3 fragments, we purified this chimeric protein of any bacterial impurities and immunized several mice to select a variant with high content of IgGs against the outer loops of each of the three *N. bombycis* ATP/ADP transporters. The representativeness of the constructed library (10^7^ transformants) was also doubled compared to the previously constructed one against the microsporidium *Vairimorpha ceranae* hexokinase [23]. Despite this, we failed to isolate any recombinant Abs specific to loops of the second and third *N. bombycis* transporters. Since *Nb*AAC2- and *Nb*AAC3-specific Abs were detected in immune serum of the mouse selected for library construction (Figure 1B), the proportion of their VH and VL transcripts in the total pool of synthesized cDNA was probably insufficient to form specific scFv Abs.

Fortunately, transcriptome analysis of *N. bombycis* mature spores and sporoplasms (microsporidia embryos infecting new host cells) [24] demonstrated that the expression of *Nb*AAC1 gene (accession number KB909159.1; 154353 FPKM (fragments per kilobase per million) for sporoplasms and 146706 ones for spores) was about 1500 and 15000 times higher than the transcription levels of *Nb*AAC2 (KB908963.1; 113 and 128 FPKM) and *Nb*AAC3 (KB909008.1; 9 and 4 FPKM, respectively) genes. This indicates that the *Nb*AAC1 transporter plays the main role in the energy supply of the first stages of the parasite intracellular development and can be considered as the most promising target for microsporidia-suppressing recombinant Abs.

Since gene sequencing of the first 10 selected Abs showed that scFv3 and 10 were identical, nine different variants were expressed in Sf9 insect cells. The presence of the recombinant c-myc peptide at the C-terminus of Abs made it possible to confirm their expression in insect cell cultures by immunoblotting before searching for a variant that maximally suppresses parasite growth and is promising for further studies. Previously, it was shown that the analysis of microsporidia transcripts involved in sporogenesis but not expressed at the initial stages of infection may be successfully used to assay the parasite growth in cell cultures, regardless of a multiplicity of their infection [25]. In the case of infection of Sf9 cells with *N. bombycis* spores, the level of expression of the gene encoding the parasite PTP2 may serve as a such marker [22]. To match the number of parasite and insect host transcripts in the same cDNA samples, we used the expression level of the *S. frugiperda* COXI gene as a reference. Mitochondrial transcripts are polyadenylated in insect cells [26], and the gene encoding COXI is highly expressed [27].

Although the formation of *N. bombycis* spores in *B. mori* ovarian tissue cultures was observed as early as 2 days after infection [28], qPCR analysis of the parasite β-tubulin gene copies [13] and immunoblotting with Abs against the same protein [22] demonstrated the parasite growth in Sf9 cells between days 4 and 7 after inoculation. This is most likely due to the immediate germination of first-generation spores to infect new insect cells. Previously, the formation of such “early” spores was described for the *Nosema* species, including *N. bombycis* [29,30,31]. With these data in mind, we analyzed infected cultures at day 7 post-infection in the case of qPCR, and samples for immunoblotting were prepared using cell cultures 1.5 h, 4 and 7 days after inoculation.

The first infection of 64 cell cultures expressing nine recombinant Abs with 20 spores per insect cell showed that (1) the number of parasite transcripts per thousand of insect ones can vary in 10 times for transformants expressing different Abs, (2) scFv5 demonstrate the most effective *N. bombycis*-inhibiting activity. Subsequent experiments with different numbers of infecting spores per insect cell, other *Nb*PTP2-2 and *Sf*COXI-2 primers, GFP-expressing control cell cultures, and immunoblotting confirmed the highest inhibitory activity of scFv5 compared to other selected variants. Interesting features of this recombinant Ab are (1) the retention of a human IgG VL fragment from the pSEX81 phagemid, (2) the unique sequence of its VH domain, including the CDRs.

Three years ago, we constructed an immune scFv library against fat body proteins of locust *Locusta migratoria* infected with the microsporidium *Paranosema* (*Antonospora*) *locustae* and selected scFv fragments specific to three secretory proteins of the parasite [32]. The following sequencing showed that two of them had the same VL domain of a human IgG (unpublished data). Recently, we also selected a recombinant Ab against the microsporidium *V. ceranae* hexokinase containing this VL domain from the commercial vector pSEX81 [23], and in this study we found it in two anti-*Vc*AAC1 scFv fragments. Some characteristics of such scFv Abs with an “artificial” VL part should be similar to those of single domain Abs (sdAbs) carrying natural VHs of camelids [33,34] and cartilaginous fishes [35,36], or to “camelized” nanobodies [37,38]. Since a high diversity of VH CDR3 loops is sufficient to provide antigen specificity of any Ab, and this region is a key determinant of antigen recognition [39], the functional role of such “artificial” VL fragment in scFv molecules may be associated with an increase in their thermal stability and a decrease in aggregation tendencies. In the case of sdAbs it is ensured by a special repertoire of hydrophilic and cysteine residues in proteins [40,41,42]. Since we have found the “artificial” VL domain in several specific recombinant Abs, its presence does not appear to be critical for the antigen-binding activity of these scFv fragments.

Along with the search for new inhibitory Abs, further modification of scFv5 should be aimed at finding its native VL fragment, which is present in a natural mouse IgG not found in the constructed library, which appears to be insufficiently representative. Cloning a pool of mouse VL fragments into the pSEX81 phagemid with previously inserted scFv5 VH part followed by panning the new library can solve this problem. Another direction may be connected with the camelization of its VH fragment, which implies the engineering introduction of additional disulfide bonds to stabilize the tertiary structure [43], hydrophilic residues, as well as randomization of CDRs and especially CDR3.

The scFv3 fragment with the similar VH domain could also be the subject of such study. Although it did not show very effective *N. bombycis* inhibitory activity, (1) it is the only recombinant Ab selected twice; (2) it showed high specificity on dot blots (Figure 2A, scFv3, 10) and the lowest level of expression in Sf9 cells (Figure 4, scFv3); (3) its VH sequence similarity to that of scFv5 suggests recognition of the same epitope; (4) its relatively extended (12 a/a) CDR3 of the VH domain (Figure 6) is not identical to any sequence annotated in GenBank. It is possible that the replacement of its VL domain with another one or engineering modifications of its VH part will help to construct scFv or sdAb that effectively suppress the intracellular growth of *N. bombycis*.

## 4. Materials and Methods

### 4.1. Bacterial Overexpression of Chimeric Proteins and Development of Immune scFv Library

The constructing of plasmids and bacterial overexpression of the *Nb*AACb chimeric protein carrying 18 outer hydrophilic domains of three *N. bombycis* ATP/ADP carriers, as well as its fragments *Nb*AAC1, *Nb*AAC2, *Nb*AAC3, consisting of loops of individual transporters, were described previously [21]. To eliminate the C-terminal tag of *Nb*AACb, the encoding sequence was amplified using its copy, previously inserted into the tagless variant of pRSETa [21] as a template, Phusion Flash High-Fidelity PCR Master Mix (Thermo Fisher Scientific, Waltham, MA, USA), forward T7 and reverse *Nb*AAC3 (Table 5) primers.

The gel-purified PCR-product was re-cloned in the tagless pRSETa plasmid at the *BamH*I/*Hind*III sites, and the *Nb*AAC1-3 devoid of any tags was overexpressed in *E. coli*, as it was previously described for other chimeric proteins [21].

### 4.2. Antigen Preparation and Immunization of Mice

To immunize mice, the *Nb*AAC1-3, forming insoluble inclusion bodies (IBs) in *E. coli*, was solubilized in the presence of SDS-PAGE sample buffer (62.5 mM Tris-Cl (pH 6.8), 2% SDS, 5% 2-mercaptoethanol, 10% glycerol) at 95 °C for 15 min, and the chimeric protein was purified by SDS-PAGE in 12% gel using a PROTEAN^®^ II xi cell (Bio-Rad, Hercules, CA, USA) followed by electroelution using the same manufacturer’s whole gel eluter. After electroelution, fractions containing pure chimeric protein were thoroughly dialyzed against TBS (50 mM Tris-Cl (pH 7.4), 150 mM NaCl) mixed with an equal volume of Freund’s adjuvant (complete for first injection, incomplete for subsequent ones) and four mice were immunized with four intraperitoneal injections (80 μg of protein per injection) at 10-day intervals. Ten days after the last immunization, sera were analyzed by immunoblotting with *Nb*AAC1-3, *Nb*AAC1, *Nb*AAC2, *Nb*AAC3 proteins transferred onto a nitrocellulose membrane. For staining of the chimeric proteins, we used immune sera, anti-mouse IgG polyclonal Abs conjugated with horseradish peroxidase (HRP) (Bio-Rad, Hercules, CA, USA) and 4-chloro-1-naphthol (Merck, Darmstadt, Germany) as a colorimetric substrate for HRP reaction [44].

### 4.3. Construction of the scFv Immune Library

For cDNA synthesis and amplification of IgG VH and VL variable domains, we used (1) RNA from mice whose serum contained Abs to the loops of all three *N. bombycis* transporters, (2) the Mouse IgG Library primer set (Progen Biotechnik, Heidelberg, Germany), (3) DreamTaq Green PCR Master mix (Thermo Fisher Scientific, Waltham, MA, USA). The gel-purified PCR products were cloned into the pSEX81 phagemid vector (Progen Biotechnik, Heidelberg, Germany) at the *Nco*I/*Hind*III (VH domains) and *Mlu*I/*Not*I (VL domains) restriction sites to construct an immune scFv library with a representativeness of about 10 million *E. coli* transformants (XL1-Blue MRF’ strain). All stages of library constructing, storage, infection with the M13 KO7ΔpIII hyperphage (Progen Biotechnik, Heidelberg, Germany), and phage production have been described previously [32].

### 4.4. Library Panning

Phages produced in 200 mL of bacterial culture were precipitated with PEG, resuspended in 3 mL of phage dilution buffer, and blocked with 4 volumes of TTBS (TBS plus 0.1% Tween-20) with 3% BSA for 30 min. *Nb*AAC1, *Nb*AAC2, and *Nb*AAC3 transferred to a nitrocellulose membrane after SDS-PAGE were cut into triangles, squares, and stripes respectively, washed in TTBS, blocked in the presence of 3% BSA for 3 h at 25 °C, placed in a 15 mL tube with page suspension and incubated overnight at 4 °C. After incubation, the nitrocellulose figurines were thoroughly washed with TTBS, TBS, and phages bound to individual chimeric proteins were separately eluted in 0.5 mL of 0.1 M triethylamine for 5 min. Aspirated eluates were immediately neutralized with the same volume of 1 M Tris-Cl (pH 7.5) and stored on ice until bacteria infecting. *E. coli* XL1-Blue MRF’ cells were infected with eluted phages as described previously [23]. Their re-infection with helper phage, PEG precipitation, incubation with immobilized antigen, elution of bound phages, followed by re-infection of bacteria allowed for the second and third rounds of selection.

### 4.5. Bacterial Expression and Analysis of Selected scFv Fragments

The re-cloning of selected scFv genes into the pOPE101 plasmid (Progen Biotechnik, Heidelberg, Germany), their expression in *E. coli*, and the analysis of the ability of selected scFv antibodies to recognize *Nb*AAC1-3, *Nb*AAC1, *Nb*AAC2, *Nb*AAC3 by dot blotting and immunoblotting were performed as described previously [23,32]. The isolation of plasmid DNA from transformants producing *Nb*AAC-specific scFv-fragments was followed by sequencing of their genes with pOPE101seq primers (Table 5) and alignment using Clustal W of MegAlign from DNASTAR’s Lasergene sequence analysis software [45]. Complementarity determining regions (CDRs) of the sequenced VH regions were predicted according to the instructions of Andrew C.R. Martin’s group at UCL (http://www.bioinf.org.uk/abs/info.html).

To express the scFv-encoding genes in insect cells, they were re-amplified with pOPEpIB primers (Table 5) and cloned into the pIB/V5-His vector at the *Sac*I/*Xho*I restriction sites. The eGFP coding gene was PCR amplified using eGFP forward and reverse primers (Table 5), Phusion Flash High-Fidelity PCR Master Mix (Thermo Fisher Scientific, Waltham, MA, USA), pEGFP-N3 plasmid (Clontech Lab Inc., San Jose, CA, USA) as a template, and cloned into the same vector at the *Eco*RI/XhoI sites. Plasmids with inserted genes were purified, verified by sequencing with pIB seq forward and reverse primers (Table 5), and used to transfect Sf9 cells.

### 4.6. Heterologous Expression of scFv Abs in Sf9 Cells and Their Infection with N. bombycis Spores

*S. frugiperda* (Lepidoptera: Noctuidae) derived Sf9 cells from the ECACC General collection (ECACC 89070101) were cultured in Sf-900III SFM (Thermo Fisher Scientific, Waltham, MA, USA) and regularly maintained according to the manufacturer’s instructions. Transformation (lipofection) of adhesion cell cultures with 10 µg of pure plasmid DNA and selection of blasticidin-resistant transformants were performed as described previously [23].

To confirm the expression of the studied scFv fragments in Sf9 cells, 5 × 10^5^ blasticidin-selected non-infected transformants or control untransformed Sf9 cells were sonicated in 0.1 mL TBS and analyzed by immunoblotting with anti-c-Myc monoclonal Abs (Merck, Darmstadt, Germany) and HRP- conjugated anti-mouse IgG polyclonal Abs (Bio-Rad, Hercules, CA, USA). ChemiDoc MP Imaging System (Bio-Rad, Hercules, CA, USA) and Clarity™ Western ECL substrate (Bio-Rad, Hercules, CA, USA) were used to detect the HRP reaction.

*N. bombycis* spores were obtained from the Uzbek Research Institute of Sericulture in Tashkent, Uzbekistan. Their (1) isolation from fat bodies of experimentally infected *Bombyx mori* 5th instar caterpillars, (2) purification in the Percoll density gradient, (3) sterilization with the antiseptic Multicide (Sante Pharm, Moscow, Russia) [46], polar tube extrusion activation [47] have also been described previously [22]. To obtain infection rates of fifty, twenty, ten and three spores per insect cell, 2.5 × 10^7^, 10^7^, 5 × 10^6^ or 1.5 × 10^6^ activated spores in 20 µL of 10 mM KOH were added to a culture plate well containing 5 × 10^5^ Sf9 cells in 500 µL of SF-900 III SFM medium (Thermo Fisher Scientific, Waltham, MA, USA). Infected cell cultures were maintained at 27 °C for 7 days without control of humidity and CO_2_ concentration. After culturing, pelleted at 600× *g* for 5 min cells were stored at −80 °C until RNA isolation, cDNA synthesis and RT-qPCR analysis.

### 4.7. qPCR Analysis of N. bombycis Growth in Infected Cultures

The isolation of total RNA using Trizol, DNAse I and RNA grade glycogen (Thermo Fisher Scientific, Waltham, MA, USA) was performed according to the manufacturer’s instructions. cDNA was synthesized in 20 μL reaction mixture containing 1 μg RNA, 1 μM oligo-dT primers, 0.6 mM dNTPs, 100 U RevertAid M-MuLV reverse transcriptase (RT), 20 U RNase inhibitor (all reagents were produced by Thermo Fisher Scientific, Waltham, MA, USA) as described previously [23].

To assay the content of *N. bombycis* polar tube protein PTP2 gene transcripts in infected transformants relative to the transcripts of the reference host gene of *S. frugiperda* COXI by real-time qPCR, cDNA, two pairs of primers *Sf*COXI/NbPTP2 or SfCOXI-2/NbPTP2-2 (Table 5) were mixed with 5× qPCRmix-HS SYBR (Evrogen, Moscow, Russia), following the manufacturer’s recommendations. For analysis, a CFX Opus real-time PCR detection system with a C1000 thermal cycler (Bio-Rad, Hercules, CA, USA) was used. Quantification cycle values (Cq, number of cycles required for fluorescence to reach a quantification threshold) were determined using the supplied software. To assess statistically significant differences between ΔCq values, one-way analysis of variance (one-way ANOVA) followed by Tukey’s test for post-hoc test (Table 2 and Table 3) or Kruskal-Wallis one-way analysis of variance followed by Dunn’s post-hoc test and relevant R functions (Table 4) were used.

To determine the number of transcripts in the analyzed samples, amplified with *Sf*COXI and *Nb*PTP2 primers (Table 5) DNA fragments were cloned into the pAL-TA vector (Evrogen, Moscow, Russia), and qPCR analysis performed on 10^5^–10^8^ or 10^2^–10^6^ plasmid copies, respectively, was followed by construction of calibration curves and counting of transcripts in samples using NEBioCalculator v1.15.0 (https://nebiocalculator.neb.com/#!/qPCRGen).

The efficiency of the *Sf*COXI and *Nb*PTP2 primers for qPCR was determined using the ThermoFisher Scientific QPCR Efficiency Calculator.

### 4.8. Analysis of N. bombycis β-Tubulin Accumulation in Infected Transformants

Sf9 cell cultures expressing selected scFv Abs were infected with 10, 20 and 50 spores per insect cell and cultured for 1.5 h, 4 and 7 days. After cultivation, they were analyzed by immunoblotting with polyclonal Abs against the parasite β-tubulin as described previously [22].

## Figures and Tables

**Figure 1 ijms-23-15307-f001:**
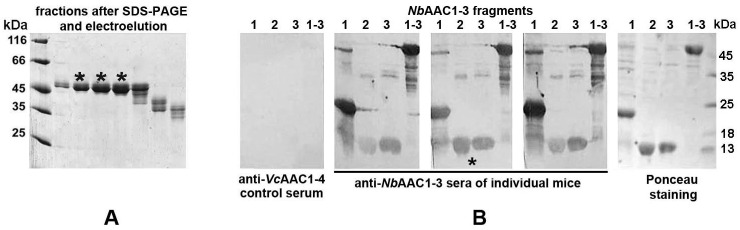
(**A**) *Nb*AAC1-3 chimeric protein carrying the outer loops of three *N. bombycis* ATP/ADP transporters was overexpressed in *E. coli*, purified by SDS-PAGE and gel electroelution. The pure protein fractions, marked with asterisks, were dialyzed and used to immunize mice. (**B**) Western blot analysis of immune sera from individual mice allowed us to select a mouse whose serum showed similar levels of Abs to the loops of all three *N. bombycis* transporters (marked with an asterisk) and use it to construct the scFv library. As a negative control, we used the serum of a mouse immunized with the chimeric *Vc*AAC1-4 protein carrying the outer loops of the microsporidium *Vairimorpha ceranae* four ATP/ADP transporters [21] also overexpressed in *E. coli* and purified.

**Figure 2 ijms-23-15307-f002:**
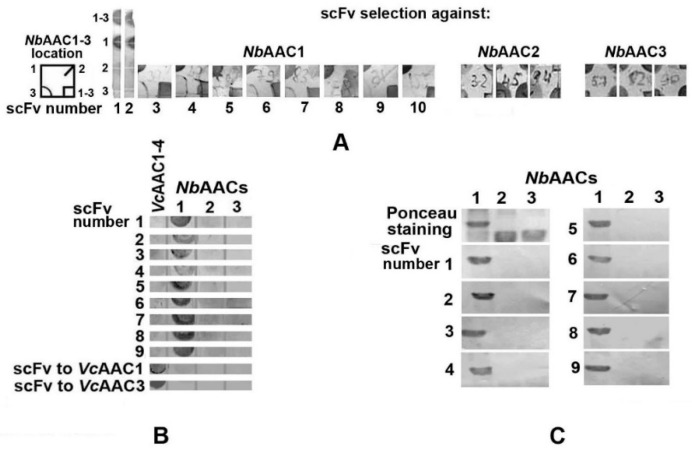
(**A**) Dot blots of 10 selected recombinant Abs showed their specific recognition of the chimeric proteins *Nb*AAC1 and *Nb*AAC1-3 but not *Nb*AAC2 and *Nb*AAC3. Dot blot analysis of some recombinant Abs selected against *Nb*AAC2 and *Nb*AAC3 demonstrated non-specific staining of three or four chimeric proteins placed at the corners of nitrocellulose squares, but none of the scFv variants specifically recognized these *Nb*AAC1-3 fragments. (**B**) As a control, we used the *Vc*AAC1-4 chimeric protein carrying the outer loops of four microsporidia *Vairimorpha ceranae* ATP/ADP transporters recognized by previously constructed *Vc*AAC1- and *Vc*AAC3-specific recombinant Abs. (**C**) Binding of selected scFv fragments to the *Nb*AAC1 protein band, but not to *Nb*AAC2 or *Nb*AAC3, was shown by immunoblotting.

**Figure 3 ijms-23-15307-f003:**
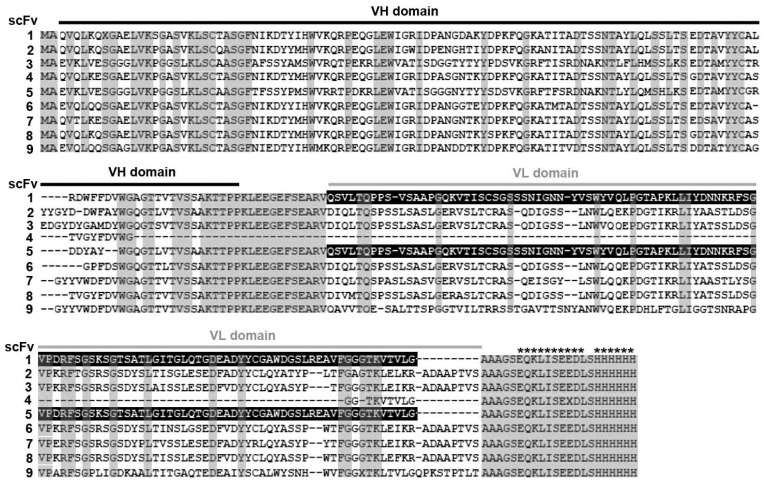
Multiple a/a sequence alignment of nine selected scFv antibodies. VH and VL domains are indicated by black and gray horizontal lines respectively, gray rectangles indicate conserved residues present in all selected scFv fragments, the human VL domain from the pSEX81 phagemid retained in scFv1 and scFv5 is marked in inverted color, recombinant c-Myc and polyHis peptides are marked with asterisks.

**Figure 4 ijms-23-15307-f004:**
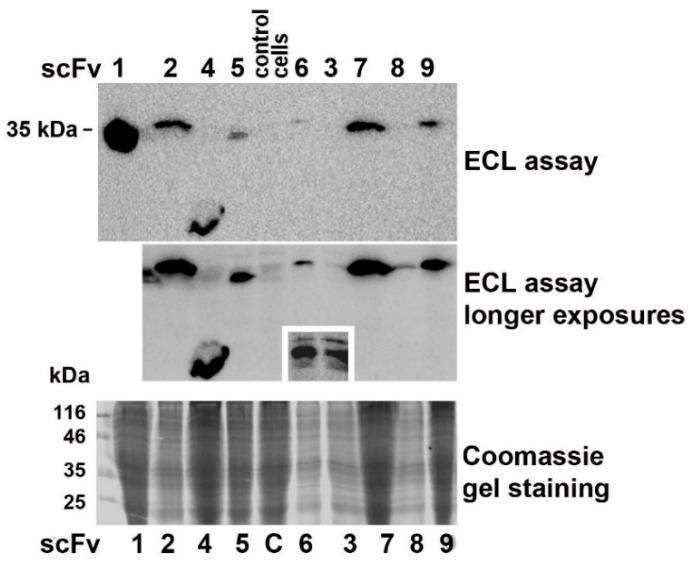
Western blotting with anti-c-myc monoclonal and HRP-conjugated anti-mouse IgG Abs confirmed expression of the nine scFv fragments in blasticidin-selected transformants. The concentration of scFv1 in the Sf9 cell culture significantly exceeded the content of other recombinant Abs. As expected, the scFv4 Ab with a single VH domain showed the lowest molecular weight, and the scFv1 and 5 fragments retaining the VL part from the pSEX81 phagemid were also smaller than the other Abs. To detect the HRP reaction, ChemiDoc MP Imaging System and Clarity™ Western ECL substrate were used. For the ECL assay of the nitrocellulose membrane, the optimal autoexposure settings of the Imaging System were used. To obtain a longer exposure, the very bright scFv1 lane was removed before the second run of analysis. To obtain the third small ECL image, we only analyzed the scFv3 and scFv6 lanes.

**Figure 5 ijms-23-15307-f005:**
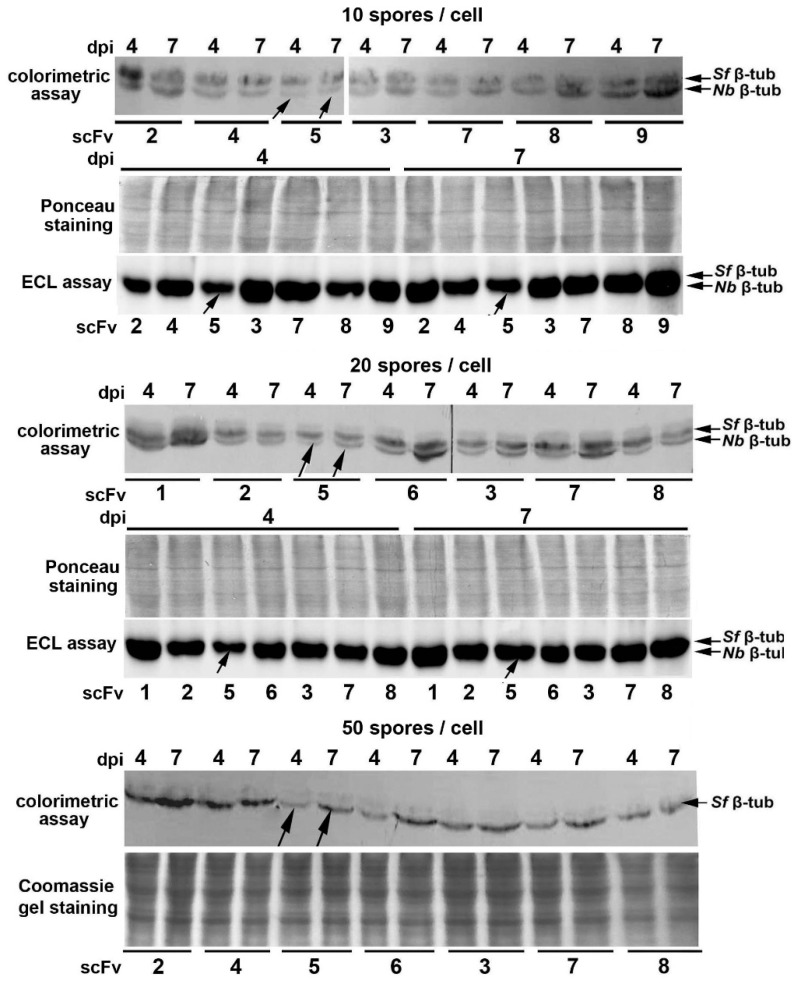
Western blot analysis with Abs against *N. bombycis* β-tubulin showed that in 6 experimental groups representing equal number of Sf9 transformants infected with three doses of spores and cultivated for 4 and 7 days, the lowest *N. bombycis* growth was observed in the case of scFv5 expression (marked by arrows). The lower detected bands corresponded to the parasite β-tubulin, and the upper one, to the *S. frugiperda* homologous protein [22]. To detect the HRP reaction, 4-chloro-1-naphthol as a colorimetric substrate or ChemiDoc MP Imaging System and Clarity™ Western ECL substrate were used.

**Figure 6 ijms-23-15307-f006:**
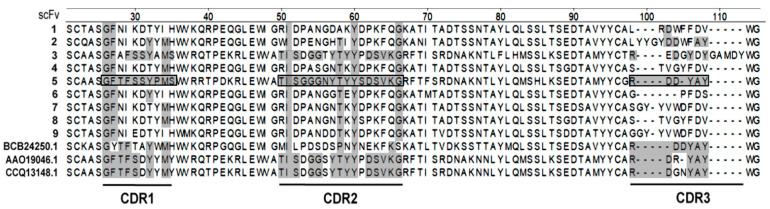
Analysis of CDRs (complementarity determining regions) of VH domains of selected recombinant Abs and three annotated in GenBank fragments with CDR3 most similar to scFv5 one. CDRs of scFv5 are framed, and identical a/a residues are marked with gray rectangles.

**Table 1 ijms-23-15307-t001:** Similarity (% identity) of the a/a sequences of the VH and VL fragments of the selected recombinant Abs.

VH	9 *	8	7	6	5	4	3	2		
** 1 **	86.2	89.7	89.7	87.6	45.6	86.2	48.3	84.5		
** 2 **	76.3	83.6	83.1	85	54.4	82.8	50.4		36.1	** 2 **
** 3 **	50	49.1	50.8	52.2	84.2	49.1		88.6	37	** 3 **
** 4 **	81.9	93.1	90.5	85	46.5		36	36.9	100	** 5 **
** 5 **	47.4	46.5	48.2	48.7		37.8	94.8	87.7	37	** 6 **
** 6 **	85.8	86.7	86.7		82.8	36.9	83.5	83.3	38	** 7 **
** * 7 * **	87.3	92.2		81	94.8	36.9	93	86.8	38	** 8 **
** 8 **	85.3		35.3	34.5	36.2	42.3	35.7	36	40.7	** 9 **
			** 8 **	** 7 **	** 6 **	** 5 **	** 3 **	** 2 **	** 1 **	**VL**

*—numbers of selected scFv Abs are in bold and underlined.

**Table 2 ijms-23-15307-t002:** Real-time qPCR analysis of *N. bombycis* PTP2 and *S. frugiperda* COXI transcripts in scFv-expressing Sf9 transformants infected with 20 microsporidia spores per insect cell and cultured for 7 days.

scFv	Cq Values * and Number of Transcripts in 1 μL of cDNA Samples (in Brackets)		ΔCq ** and Number of *Nb*PTP2 Transcripts per 10^3^ *Sf*COXI Ones (in Brackets)	N Samples and Variants with ΔCq ≥ 7 (in Brackets)
	*Sf*COXI	*Nb*PTP2		
1	13.8 ± 0.2 (7.1 × 10^6^)	20.9 ± 0.6 (20.7 × 10^3^)	7.1 ± 0.6 ^a^ (2.9)	9 (6)
2	14.2 ± 0.3 (5.4 × 10^6^)	21.6 ± 0.5 (12.3 × 10^3^)	7.4 ± 0.6 ^a^ (2.3)	9 (4)
3	15.3 ± 0.5 (2.7 × 10^6^)	22.2 ± 0.4 (7.9 × 10^3^)	6.9 ± 0.2 ^a^ (3)	6 (3)
4	14.7 ± 0.2 (3.9 × 10^6^)	19.9 ± 0.7 (43.3 × 10^3^)	5.2 ± 0.6 ^b^ (11)	5 (1)
5	13.5 ± 0.3 (8.6 × 10^6^)	22 ± 0.3 (9.2 × 10^3^)	8.5 ± 0.5 ^a^ (1.1)	8 (8)
6	14.3 ± 0.2 (5.1 × 10^6^)	20.1 ± 0.4 (37. 4 × 10^3)^	5.8 ± 0.4 ^b^ (7.3)	9 (2)
7	14 ± 0.3 (6.2 × 10^6^)	21.8 ± 0.3 (10.6 × 10^3)^	7.8 ± 0.5 ^a^ (1.7)	4 (4)
8	14.2 ± 0.5 (5.4 × 10^6^)	20.2 ± 0.4 (34.7 × 10^3^)	6 ± 0.6 ^b^ (6.4)	8 (1)
9	14.7 ± 0.3 (3.9 × 10^6^)	22.7 ± 0.5 (5.4 × 10^3^)	8 ± 0.3 ^a^ (1.4)	6 (6)

*—mean of values from 4–9 independent infections (*n* = 12–27) ± SE; **—different letters indicate statistically significant differences (*p*-value < 0.05) between scFv variants as analyzed by one-way analysis of variance (one-way ANOVA) followed by Tukey’s test for post-hoc analysis.

**Table 3 ijms-23-15307-t003:** Real-time qPCR analysis of *N. bombycis* PTP2 and *S. frugiperda* COXI transcripts in 7-day old Sf9 cultures expressing four recombinant Abs and infected with ten and three microsporidia spores per insect cell.

scFv	Spores per Cell	Cq Values * and Number of Transcripts in 1 μL of cDNA Samples (in Brackets)		ΔCq ** and Number of *Nb*PTP2 Transcripts per 10^6^ *Sf*COXI Ones (in Brackets)
		*Sf*COXI	*Nb*PTP2	
1	10	15.5 ± 0.4 (2.3 × 10^6^)	23.2 ± 0.2 (3.8 × 10^3^)	7.7 ± 0.5 ^a^ (1.7 × 10^3^)
5	10	14.4 ± 0.2 (5.1 × 10^6^)	25.1 ± 0.3 (1.2 × 10^3^)	10.7 ± 0.4 ^b^ (2.4 × 10^2^)
7	10	14.8 ± 0.2 (3.7 × 10^6^)	23.7 ± 0.2 (2.6 × 10^3^)	8.9 ± 0.2 ^a^ (7 × 10^2^)
9	10	13.8 ± 0.3 (7.1 × 10^6^)	22.1 ± 0.3 (8.5 × 10^3^)	8.3 ± 0.9 ^a^ (1.2 × 10^3^)
1	3	14.7 ± 0.1 (3.9 × 10^6^)	22.4 ± 0.1 (6.8 × 10^3^)	7.7 ± 0.2 ^a^ (1.7 × 10^3^)
5	3	15.5 ± 0.3 (2.3 × 10^6^)	26.1 ± 0.2 (4.4 × 10^2^)	10.6 ± 0.2 ^b^ (1.9 × 10^2^)
7	3	15.3 ± 0.6 (2.7 × 10^6^)	24.1 ± 0.9 (1.9 × 10^3^)	8.8 ± 0.4 ^a^ (7 × 10^2^)
9	3	14.9 ± 0.2 (3.4 × 10^6^)	23.9 ± 0.2 (2.2 × 10^3^)	9 ± 0.3 ^a^ (6.5 × 10^2^)

*—mean of values from three independent infections (*n* = 9) ± SE; **—different letters indicate statistically significant differences (*p*-value < 0.05) between ΔCq values in scFv5 and other samples as analyzed by one-way analysis of variance (one-way ANOVA) followed by Tukey’s test for post-hoc analysis.

**Table 4 ijms-23-15307-t004:** Real-time qPCR analysis of *N. bombycis* PTP2 and *S. frugiperda* COXI transcripts with primers *Sf*COXI-2 and *Nb*PTP2-2 in Sf9 cultures expressing five *Nb*AAC1-specific recombinant Abs and eGFP (control). Blasticidin-selected transformants were infected with 50 microsporidia spores per insect cell and incubated 7 days prior to assay.

scFv	Cq *		ΔCq **
	*Sf*COXI-2	*Nb*PTP2-2	
2	19.4 ± 0.1	21.7 ± 0.1	2.3 ± 0.1 ^a^
3	19.7 ± 0.8	22.1 ± 0.4	2.4 ± 0.5 ^a^
5	20.6 ± 0.4	26.4 ± 1	5.8 ± 0.6 ^b^
7	18.1 ± 0.3	20.4 ± 0.4	2.3 ± 0.4 ^a^
8	19.2 ± 0.1	19.6 ± 0.1	0.4 ± 0.1 ^a^
eGFP	18.5 ± 0.2	19.1 ± 0.2	0.6 ± 0.2 ^a^

*—mean of values from 2 independent infections (*n* = 6) ± SE; **—different letters indicate statistically significant differences (*p*-value < 0.05) between scFv variants as analyzed by Kruskal-Wallis test followed by Dunn’s post-hoc test.

**Table 5 ijms-23-15307-t005:** List of primers used for PCR amplification and sequencing of *N. bombycis*, *S. frugiperda*, scFv genes and their fragments.

Primer	Sequence (5′-3′)
*Nb*AAC3 rev ^a^	TGCAAGCTTACGGTGCAGCACGACGAATGTTGTC
pOPE101seq for	AAGAGGAGAAATTAACCATGA
pOPE101seq rev	TCATTAGCACAGGCCTCTAGA
pOPEpIB for ^b^	ACGGAGCTCGGTACCTGCTGCTGGCAGCTCAGCCGGCCATG
pOPEpIB rev ^c^	GCTGAATTCTCGAGTTAATGATGATGGTGATGATGGGATAG
GFP for ^d^	CATGAATTCATGGTGAGCAAGGGCGAGGAGCTG
GFP rev ^e^	TCACTCGAGTTACTTGTACAGCTCGTCCATGCCGA
pIB seq for	CGCAACGATCTGGTAAACAC
pIB seq rev	GACAATACAAACTAAGATTTAGTCAG
*Sf*COXI for	TTTGAGCAGGAATAGTAGGT
*Sf*COXI rev	TAAAGATGGGGGTAAAAGT
*Sf*COXI-2 for	TACCGCATTTTTATTATTATTATC
*Sf*COXI-2 rev	GTTTCCTTTTTACCTCTTTCTTGA
*Nb*PTP2 for	TGGCATCAGTAGCTCCTCCTCAAG
*Nb*PTP2 rev	ACGGCCCTAGCTGCTGTTTCAA
*Nb*PTP2-2 for	CAATAATCCAGCCGAGTGTCAA
*Nb*PTP2-2 rev	AGTGGGGTACCTTCAGCAGTTT

^a^—*Hind*III site and stop codon are underlined; ^b^—*Sac*I/*Kpn*I sites are underlined; ^c^—*Eco*RI/*Xho*I and stop codon are underlined; ^d^—*Eco*RI site is underlined; ^e^—*Xho*I site and stop codon are underlined.

## Data Availability

Not applicable.

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
