# Peer review of "Construction of scFv Antibodies against the Outer Loops of the Microsporidium Nosema bombycis ATP/ADP-Transporters and Selection of the Fragment Efficiently Inhibiting Parasite Growth"

_ijms, 2022, doi:10.3390/ijms232315307_

Round 1

Reviewer 1 Report

Comments and Suggestions for Authors

Pébrine is a serious microsporidiosis in sericulture. How defense this disease is an important subject in sericulture. In previous research, authors constructed the antigen through fusing 18 outer hydrophilic loops of three N. bombycis AACs. Here, authors constructed a mouse scFv library based on previous work and screened the fragment which most effectively inhibited the parasite growth. The inhibitory efficiency was demonstrated in sf9 cells. This work made a meaningful exploration for constructing transgenic silkworm strains with microspore resistance. But I still have some concerns and suggestion.

Major comments:

1.      The author verified effectiveness of scFv Abs in sf9 cell. Why didn’t do it in BmN Cell which derived from ovarian of silkworm? BmN cells maybe more suitable to verify the suppress effectiveness of scFv5, and it also can test whether the Abs has nonspecific binding in silkworm.

2.      In table 2 and table3, author compared suppress intracellular growth of N. bombycis between different Abs. However, my concern is that there was no blank control. Compared with the cells without expressing Ab, how suppress effectiveness is?

3.      What’s the internal reference (IR) in your western-blotting experiment? Please provide the normalization image of IR. I ask this question because the quantity of Sf β-tubulin are also different between samples. Although author claimed that the lowest growth of the pathogen was observed in the case of scFv5 expression, the staining intensity of the Sf β-tubulin in that samples were also lower. 

Minor comments:

1.     In line 123, “10 selected recombinant Abs (Table 1, Figure 3)”. However, 9 scFv Abs were listed in table 1 and figure 3. Please confirm. Meanwhile, the bold words don’t stand out in table1. The underline may be better.

2.     I suggestion that table 2 and 3 can be converted to bar plot. It will be more intuitive and readable.

3.     The amplifition efficiency of primers for qRT-PCR should be provided.

Author Response

1st Reviewer

Comments and Suggestions for Authors

Pébrine is a serious microsporidiosis in sericulture. How defense this disease is an important subject in sericulture. In previous research, authors constructed the antigen through fusing 18 outer hydrophilic loops of three N. bombycis AACs. Here, authors constructed a mouse scFv library based on previous work and screened the fragment which most effectively inhibited the parasite growth. The inhibitory efficiency was demonstrated in sf9 cells. This work made a meaningful exploration for constructing transgenic silkworm strains with microspore resistance. But I still have some concerns and suggestion.

Major comments:

1.The author verified effectiveness of scFv Abs in sf9 cell. Why didn’t do it in BmN Cell which derived from ovarian of silkworm? BmN cells maybe more suitable to verify the suppress effectiveness of scFv5, and it also can test whether the Abs has nonspecific binding in silkworm.

Sf9 is one of four Lepidoptera cell lines recommended by Invitrogen for the expression of foreign proteins using the pIB/V5-His vector: “The virus’ natural host is the Douglas fir tussock moth; however, the promoters allow protein expression in Lymantria dispar (LD652Y), Spodoptera frugiperda cells (Sf9) (Hegedus et al., 1998; Pfeifer et al., 1997), Sf21 (Invitrogen), Trichoplusia ni (High Five™) (Invitrogen)” (User manual InsectSelect™ BSD System For Stable Expression of Heterologous Proteins in Lepidopteran Insect Cell Lines using pIB/V5-His).  Thus, we have used this reliable expression system. In addition, previous expression of scFv Abs suppressing the microsporidia growth was also performed in this cell line [13,14, 23].

  1. In table 2 and table3, author compared suppress intracellular growth of N. bombycis between different Abs. However, my concern is that there was no blank control. Compared with the cells without expressing Ab, how suppress effectiveness is?

For the primary screening of the constructed and selected scFv variants in the first and second qPCR experiments (Tables 2 and 3) and in immunoblotting (Fig. 5), transformants with more efficient growth of the parasite can be considered as such a control. They are most similar to cells expressing inhibitory scFv. Since the presence of the antibiotic blasticidin in the medium and the effective accumulation of scFv molecules in Sf9 transformants (Fig. 4, scFv1, 2) can affect the growth of the parasite without direct interaction with its ATP/ADP carriers, such controls are most accurate at the first stage of the study. The third qPCR experiment (Table 5) can be seen as the start of a more detailed analysis of scFv5 using various standard controls, and we infected GFP-expressing transformants as the first one.

  1. What’s the internal reference (IR) in your western-blotting experiment? Please provide the normalization image of IR. I ask this question because the quantity of Sf β-tubulin are also different between samples. Although author claimed that the lowest growth of the pathogen was observed in the case of scFv5 expression, the staining intensity of the Sf β-tubulin in that samples were also lower. 

In the new MS version Fig.5 was modified: (1) We repeated immunoblotting of samples of cultures infected with 10 and 20 spores per insect cell and combined them in 4 dpi and 7 dpi groups. (2) Proteins transferred to nitrocellulose were stained with Ponceau, to show their similar content in these samples. (3) These membranes were stained with the more sensitive ECL method to show the same intensity of decoration of the upper part of the double band corresponding to Sf b-tubulin. (3) Since only the Nb b-tubulin band was predominantly stained in the case of infection 50 spores/cell, we simply repeated SDS-PAGE analysis followed by Coomassie staining to confirm similar protein concentration in these samples.

Minor comments:

  1. In line 123, “10 selected recombinant Abs (Table 1, Figure 3)”. However, 9 scFv Abs were listed in table 1 and figure 3. Please confirm. Meanwhile, the bold words don’t stand out in table1. The underline may be better.

First, we selected 10 NbAAC1-specific Abs by the constructed library panning. However, sequencing of their genes demonstrated that scFv3 and 10 were identical. Thus, 9 scFv fragments were analyzed in further experiments. We modified Table 1 in the new MS version.

  1. I suggestion that table 2 and 3 can be converted to bar plot. It will be more intuitive and readable.

Although the graphical representation of the data is indeed more readable, in this case we should limit such a representation to ΔCq or number of NbPTP2 transcripts per 103 SfCOXI ones. At the same time, columns 2 and contain 3 some interesting information (similar SfCOXI Cq values in all analyzed cultures, the ability to compare NbPTP2 Cq values in different cells regardless of the SfCOXI Cq ones etc.). In our opinion, the presentation of such quantitative data in tables is more informative for their thoughtful analysis.

  1. The amplification efficiency of primers for qRT-PCR should be provided.

We have added the following phrase to Results: “The constructing of calibration curves also showed that PCR efficiencies was 92.4% and 122.4% for primers SfCOXI and NbPTP2, respectively.” and in M&M: "The efficiency of the SfCOXI and NbPTP2 primers for qPCR was determined using the ThermoFisher Scientific QPCR Efficiency Calculator."

Reviewer 2 Report

Construction of scFv Antibodies Against the Outer Loops of the Microsporidium Nosema bombycis ATP/ADP-Transporters and Selection of the Fragment Efficiently Inhibiting Parasite Growth

Viacheslav V. Dolgikh et al

submitted for Int J Mol Sci

Review of the manuscript

Recommended for acceptance and publication

The manuscript tenders a new output on developing antibody against growth of the potential parasite Nosema bombycis. The authors exercised strong experimental support to develop the antibody scFv against the antigen ATP/ADP-Transporters of N. bombycis. The MS is well presented and well written, though I suggest English language edition before acceptance.

Following suggestions for improvement have been given which can be considered for minor revision

Figure 3: Preferred multiple seq alignment of the amino acid sequences of scFv1with scFv3 and scFV10 to show the similarity and sequence of VL of human IgG. (Fig. 6 is a similar study however similarity is lower)

Figure 4: Pl give parallel house keeping / COX in blot

What is the insert in Figure 4? please give in legend (after long exposure)

Exposure time in both normal and long term blots?

Figure 5: legend…the lowest N. bombycis growth was observed in the case of scFv5

expression (white arrows). The lower detected bands corresponded to the parasite β-tubulin

Make a correlation analysis between Western blot intensity with gene expression?...in order to show the lowest N. bombycis growth?

There can be age-wise growth difference. Moreover cytoskeletal protein tubulin could be expressed high in early ages where as PTP2 showed an increase in expression on day 7 (later age) (Tables 2 & 3 ) probably due to requirement of release of PTP for infection of new cells. However in the presence of antibody, scFv5, Nosema  growth was inhibited, illustrated by low PTP presence.

Single cell sorter (FACs) analysis is preferred here to show the level of scFv5 after inducing cell permeability.

Lines 372-373 and 381-382 showed structure of the Ab VH / VL region where cystein residues could be involved in disulphide bond formation.

As disulphide bonds could stabilize the tertiary structure (McAuley A, et al Protein Sci. 2008 Jan;17(1):95-106), the discussion on the Ab structure may be clarified by restructuring these statements.

Author Response

2nd Reviewer

  1. The last paragraph of the introduction section should be split into two sections, one regarding to the background info of the transporter protein, the other should be a highly summarized intro about this manuscript.

In the new MS version, the last paragraph begins with the phrase " Previously, we fused 18 outer hydrophilic loops of three N. bombycis AACs (NbAACs) contacting with the cytoplasm of an infected host cell [21].”

  1. Result 2.1, This paragraph will need further necessary interpretation which should be brief, to make it more readable to the readers. Such as some brief info regarding how the antigen was constructed and how the animal was immunized. Although these above info might have been described in other places, the current info disclosed in this paragraph is not sufficient and a little bit confused.

We have reduced the first part of Results (2.1), removing all considerations about the different immunogenicity of NbAAC1-3 fragments. In the new MS version, 2.1 and 2.2 are combined under the title “2.1. Constructing of scFv immune library, its panning and selection of NbAAC1-specific Abs”

  1. Fig 1B

(1) the western assay need to be redone to make the point. The current quality of the results is just not accepted.

We repeated WB with sera from three immunized mice on the same gel and added a negative control.

(2) if the authors would like to make the point by comparing the intensity among different antigens, each sample at least need to be run on the same gel and subjected to the sample following procedure. In addition, there are really many possible reasons to explain the observation, such as the loading amount of the sample. There are not sufficient info in the context, e.g., in method section, to help to support the authors’ point.

We omitted the assumptions about the different immunogenicity of chimeric proteins and the intensity of their staining in the new version of MS.

(3) for such assay using poly sera, at least a negative control will be needed.

To repeat the WB, we used the serum from a mouse immunized with the VcAAC1-4 chimeric protein carrying the outer loops of the microsporidium Vairimorpha ceranae four ATP/ADP transporters also overexpressed in E. coli and purified.

(4) WB assay alone is not suitable assay here to make such decision, will suggest combining with such as ELISA.

We agree that our decision is not supported by experimental data.

  1. Fig 1A, Need further annotation about what samples in each lane. There is a typo with the unit of the ladder.

“Fractions after SDS-PAGE and electroelution” was added at the top of Fig.1A, and the corresponding legend has also been expanded: “Figure 1. (A) NbAAC1-3 chimeric protein carrying outer loops of three N. bombycis ATP/ADP transporters was overexpressed in E. coli, purified by SDS-PAGE and gel electroelution. Pure protein fractions, marked with asterisks, were dialyzed and used to immunize mice.”  The unit of the ladder was also corrected.

 5."Line 113-116, how the authors explain the recognition of sera against target proteins in Fig 1B, but not here?

We discuss this issue in the first paragraph of Discussion (the first sentence has been modified):

“To construct an immune library of scFv Abs specific to NbAAC1-3 fragments, we purified this chimeric protein of any bacterial impurities and immunized several mice to select a variant with high content of IgGs against the outer loops of each of the three N. bombycis ATP/ADP transporters. The representativeness of the constructed library (107 transformants) was also doubled compared to the previously constructed one against the microsporidium Vairimorpha ceranae hexokinase [23]. Despite this, we failed to isolate any recombinant Abs specific to loops of the second and third N. bombycis transporters. Since NbAAC2- and especially NbAAC3-specific Abs were detected in immune serum of the mouse selected for library construction (Figure 1B), the proportion of their VH and VL transcripts in the total pool of synthesized cDNA was probably insufficient to form specific scFv Abs.”

  1. Fig 2, The quality of these data is really not good enough for publication. Suggest to re-run. Not sure any particular reason or preference why the authors would like to run the dot blot in this way. But what you could do is either using the dot blot instrument, or if not having one, our way usually is to mark the NC membrane with lines, then load the samples using multichannel pippet on the NC membrane in each line, dry the membrane in the air for ~5 min, then proceed with the following incubation steps, whatever that would be.

We use this unusual approach for large-scale screening of 100 or more scFv-producing bacterial transformants. This allows (1) to get four such squares by applying the antigen in one spot, (2) to apply up to five different antigens per one square (four in the corners and one in the center). Then we place such squares in the wells of a 96-well plate for incubation with scFv Abs. Although they look unusually, I would prefer to keep them in Figure 2A to demonstrate the successful primary screening of scFv Abs against NbAAC1 and the unsuccessful ones against NbAAC2 and NbAAC3. Since in the last cases Abs were unspecific, they are not stored and we cannot repeat this experiment.

To confirm the specificity of the selected scFv Abs in Figure 2B, we applied fragments of the chimeric protein NbAAC1-3, as well as the control chimeric protein VcAAC1-4, carrying the outer loops of four ATP/ADP transporters of the microsporidium V. ceranae (pathogen of bees)  onto nitrocellulose strips, cut  them in half and incubated with nine selected antibodies, and also with previously obtained scFv fragments to the loops of the first (VcAAC1) and third (VcAAC3) V. ceranae transporters. Finally, we performed immunoblotting (Figure 2C) to show binding of selected Abs to the NbAAC1 protein band, but not to the NbAAC2 and NbAAC3 proteins.

  1. Table 2, need to show negative control (without any scFv expression).

For the primary screening of the constructed and selected scFv variants in the first and second qPCR experiments (Tables 2 and 3) and in immunoblotting (Fig. 5), transformants with more efficient growth of the parasite can be considered as such a control. They are most similar to cells expressing inhibitory scFv. Since the presence of the antibiotic blasticidin in the medium and the effective accumulation of scFv molecules in Sf9 transformants (Fig. 4, scFv1, 2) can affect the growth of the parasite without direct interaction with its ATP/ADP carriers, such controls are most accurate at the first stage of the study. The third qPCR experiment (Table 5) can be seen as the start of a more detailed analysis of scFv5 using various standard controls, and we infected GFP-expressing transformants as the first one.

Reviewer 3 Report

1. The last paragraph of the introduction section should be split into two sections, one regarding to the background info of the transporter protein, the other should be a highly summarized intro about this manuscript.

2. Result 2.1, This paragraph will need further necessary interpretation which should be brief, to make it more readable to the readers. Such as some brief info regarding how the antigen was constructed and how the animal was immunized. Although these above info might have been described in other places, the current info disclosed in this paragraph is not sufficient and a little bit confused.

3. Fig 1B

(1) the western assay need to be redone to make the point. The current quality of the results is just not accepted.

(2) if the authors would like to make the point by comparing the intensity among different antigens, each sample at least need to be run on the same gel and subjected to the sample following procedure. In addition, there are really many possible reasons to explain the observation, such as the loading amount of the sample. There are not sufficient info in the context, e.g.,  in method section, to help to support the authors’ point.

(3) for such assay using poly sera, at least a negative control will be needed.

(4) WB assay alone is not suitable assay here to make such decision, will suggest combining with such as ELISA.

4. Fig 1A, Need further annotation about what samples in each lane. There is a typo with the unit of the ladder.

5. Line 113-116, how the authors explain the recognition of sera against target proteins in Fig 1B, but not here?

6. Fig 2, The quality of these data is really not good enough for publication. Suggest to re-run. Not sure any particular reason or preference why the authors would like to run the dot blot in this way. But what you could do is either using the dot blot instrument, or if not having one, our way usually is to mark the NC membrane with lines, then load the samples using multichannel pippet on the NC membrane in each line, dry the membrane in the air for ~5 min, then proceed with the following incubation steps, whatever that would be.

7. Table 2, need to show negative control (without any scFv expression).

Author Response

3rd Reviewer

The manuscript tenders a new output on developing antibody against growth of the potential parasite Nosema bombycis. The authors exercised strong experimental support to develop the antibody scFv against the antigen ATP/ADP-Transporters of N. bombycis. The MS is well presented and well written, though I suggest English language edition before acceptance.

Following suggestions for improvement have been given which can be considered for minor revision

Figure 3: Preferred multiple seq alignment of the amino acid sequences of scFv1with scFv3 and scFV10 to show the similarity and sequence of VL of human IgG. (Fig. 6 is a similar study however similarity is lower)

In the new MS version, we have modified Figure 3 by performing multiple alignment of scFv amino acid sequences and marking (1) conserved residues present in all 9 sequences, (2) VH and VL domains, (3) identical human VL fragments of scFv 1 and 5, (4) recombinant c-Myc and polyHis peptides. The absence of VL domain in scFv 4 also is seen.

Figure 4: Pl give parallel house keeping / COX in blot

We modified Figure 4 and added the next phrases in the text: “To correlate scFv accumulation in the transformants with the total protein concentrations in the samples, we repeated SDS-PAGE with the same aliquots and stained the gel with Coomassie. Since the lowest content of scFv 3, 6 and 8 in transformants correlated with lower protein concentrations in these samples, this generally indicates efficient expression of recombinant Abs in Sf9 cells.”

What is the insert in Figure 4? please give in legend (after long exposure)

Exposure time in both normal and long term blots?

We have added the next phrase in Figure 4 legend: “For the ECL assay of the nitrocellulose membrane, the optimal autoexposure settings of the Imaging System were used. To obtain a longer exposure, the very bright scFv1 lane was removed before the second run of analysis. To get the third small ECL image, we only analyzed the scFv3 and scFv6 lanes”.

Figure 5: legend…the lowest N. bombycis growth was observed in the case of scFv5expression (white arrows). The lower detected bands corresponded to the parasite β-tubulin

Make a correlation analysis between Western blot intensity with gene expression?...in order to show the lowest N. bombycis growth?  

Visual assessment of protein band staining after immunoblotting is not quantitative, and the correlation between qPCR and WB data is not perfect. For example, scFv 4 showed the lowest NbPTP2 Cq values ​​as well as delta Cq in qPCR experiments, and efficient growth of the parasite in scFv4-expressing culture infected with 50 spores per cell was confirmed by immunoblotting. In cells infected with 10 spores per cell, Western blot showed more efficient growth of the parasite in the case of scFv4 expression compared to scFv5 expression, but not the most intensive among the variants studied. Nevertheless, the results of immunoblotting confirm the promise of scFv5 for further research, since none of the six groups contained the scFv fragment more effectively suppressing the development of the pathogen than this variant.

There can be age-wise growth difference. Moreover cytoskeletal protein tubulin could be expressed high in early ages where as PTP2 showed an increase in expression on day 7 (later age) (Tables 2 & 3) probably due to requirement of release of PTP for infection of new cells. However in the presence of antibody, scFv5, Nosema growth was inhibited, illustrated by low PTP presence.

Previously, we showed that, in contrast to N. bombycis b-tubulin, SWP 25,30,32, and PTP1 genes, PTP2 transcripts were absent in cultures at the early stage of infection (1.5 h post infection) [22]. To be independent of multiplicity of infection, we used PTP2-specific primers in this study. The question of the aging of the “parasite population” in infected cell cultures is indeed very important and interesting, and we discussed it in the MS: “Although the formation of N. bombycis spores in Bombyx mori ovarian tissue cultures was observed as early as 2 days after infection [28], qPCR analysis of the parasite b-tubulin gene copies [13] and immunoblotting with Abs against the same protein [22] demonstrated the parasite growth in Sf9 cells between days 4 and 7 after inoculation.  This is most likely due to the immediate germination of first-generation spores to infect new insect cells…[29-31].” Additionally, cultivation of infected cultures in the presence of the antibiotic blasticidin as well effective accumulation of scFv fragments in host cell (Fig 4, scFv 1 and 2) may somehow shift this process. That is why we considered the transformants showing ineffective microsporidia suppression as the most similar and accurate controls to select highly effective microsporidia-suppressing scFv molecules.

Single cell sorter (FACs) analysis is preferred here to show the level of scFv5 after inducing cell permeability.

It would be very interesting to use such an approach. Thanks a lot for the good idea.

Lines 372-373 and 381-382 showed structure of the Ab VH / VL region where cystein residues could be involved in disulphide bond formation. As disulphide bonds could stabilize the tertiary structure (McAuley A, et al Protein Sci. 2008 Jan;17(1):95-106), the discussion on the Ab structure may be clarified by restructuring these statements.

Thank you very much, this reference was added to Discussion: “Another direction may be connected with the camelization of its VH fragment, which implies the engineering introduction of additional disulfide bonds to stabilize the tertiary structure [43], hydrophilic residues, as well as randomization of CDRs and especially CDR3.”

  1. McAuley, A.; Jacob, J.; Kolvenbach, C.G.; Westland, K.; Lee, H.J.; Brych, S.R.; Rehder, D.; Kleemann, G.R.; Brems, D.N.; and Matsumura M. Contributions of a disulfide bond to the structure, stability, and dimerization of human IgG1 antibody CH3 domain. Protein Sci. 2008, 17, 95–106.

Round 2

Reviewer 1 Report

I think the authors have been resolved all my questions carefully. I have no question. I am looking forward to seeing the authors’ further work would be translated into actual production. 

Reviewer 3 Report

All my comments have been responded.

Thank you!